# Examining representation of women in leadership of professional medical associations in India

**Pratishtha Singh**[1], **Veena Sriram**[2,3]*, **Sonali Vaid**[4], **Sharmishtha Nanda**[5], **Vikash R. Keshri**[1,6,7]

**1** The George Institute for Global Health, New Delhi, India, **2** School of Population and Public Health, University of British Columbia, Vancouver, BC, Canada, **3** School of Public Policy and Global Affairs, University of British Columbia, Vancouver, BC, Canada, **4** Inclue Labs, New Delhi, India, **5** Independent Consultant, India, **6** School of Population Health, Faculty of Medicine and Health, University of New South Wales, Sydney, Australia, **7** State Health Resource Centre, Raipur, India

* veena.sriram@ubc.ca

**Data Availability Statement:** All data used in the study is publicly available from the websites of Professional Medical Associations. The compiled data is available as Supplementary file 2.

## Abstract

Women constitute 70% of the global health workforce but are significantly underrepresented in leadership positions. In India, professional medical associations (PMAs) play a crucial role in shaping policy agenda in the health sector, but very little is known about gender diversity in their leadership. Therefore, we analysed the gender representation of current and past leaderships of Indian PMAs. Data of the current and past national leadership and leadership committees of 46 leading PMAs representing general, specialities, and super-specialities were extracted from their official websites. Gender composition of leadership was analysed using a sequential approach. For Indian Medical Association (IMA), the largest Indian PMA, an analysis of its 32 sub-chapters was also undertaken. The findings revealed that only 9 (19.5%) out of 46 associations are currently led by a woman. Leadership committees of half the associations have less than 20% women, while there were no women in the central committee of nine PMAs. Among past presidents, information was publicly available for 31 associations and all of them have had less than 20% of women presidents till date. Among the 64 individuals currently serving as presidents and secretaries of 32 sub-chapters of IMA, only three (4.6%) are women. Even in associations closely related to women's health, such as obstetrics and gynecology, pediatrics, and neonatology, unequal representation persists, highlighting male dominance. These results demonstrate significant gender disparities in PMA leadership in India, necessitating urgent efforts to promote gender equality. Gender-transformative leadership is crucial to develop gender-sensitive health care policies and practices which can serve as a catalyst for broader societal change.

## Introduction

Men constitute only 30% of the overall health workforce yet occupy most of the leadership positions in health-related organizations [1]. On the contrary, women constitute around 70%

**Funding:** The authors received no specific funding for this work.

**Competing interests:** Three of the authors declare no competing interests. VS is on the Editorial Board of PLOS Global Public Health. VRK is a life member of the Indian Association of Preventive and Social Medicine (IAPSM) and Indian Public Health Association (IPHA) and has voting rights to elect leadership committees in these.

of the workforce in the health and social sector globally and yet they are mostly concentrated in lower-paid cadres, undertake the majority of skilled and unpaid work, and are under-represented in leadership positions [1–4]. Historically, the representation of women in the health sector exhibits an inverse pyramid pattern, characterized by a larger proportion at the base of the organizational hierarchy, while only a limited number occupy leadership positions [5]. Recent reports indicate that the proportion of women in leadership positions has remained unchanged over the past five years (2018–2022), with only 25% occupying senior leadership roles globally [3, 4]. This gendered segmentation of the global health workforce has been attributed to a variety of individual, interpersonal, institutional, and community-level barriers that limit their participation in leadership [1, 3].

In India, women constitute 29% of allopathic doctors and 80% of nurses but occupy only around 28% of leadership roles across national health organizations [3, 6]. Since 1886, when Anandibai Joshi became the first woman to graduate with a degree in Western medicine, women have played an indelible role in the development of the profession [7]. In recent times, the number of women enrolling in medical education has consistently been equal to or higher than men, yet the number of women in leadership positions in the health sector remains disproportionately low [8]. This gender disparity profoundly influences the development, implementation, and allocation of health policies and programs, impacting their equitable distribution among populations [6, 9]. It has been well established that diversity, equity, and inclusion facilitates the emergence of ideas and proposals through a broader range of perspectives that contribute towards more equitable approaches to solving health challenges [3, 10]. Beyond its impacts on health policy, the lack of gender representation has major implications for systemic discrimination and marginalization of women in the health workforce. In recent years, gender diversity in the global health workforce and leadership has been extensively debated, especially the gender imbalance in editorial roles of medical academic journals, global health organization leadership and global conferences [6, 11–19].

Professional medical associations (PMAs) are voluntary organizations of diverse groups of medical professionals by dimensions such as specialty or topical focus, career stage, or across these dimensions, as in the case of 'umbrella' associations such as national medical associations [20, 21]. In India, the number of PMAs have grown in tandem with the growth and recognition of medical specialities. These associations are known to command significant influence in shaping healthcare agendas and policies at the national and subnational level by collaborating with and/or influencing governments and other stakeholders. The role of PMAs in India range from training, development of standards and guidelines, consultations, direct service provision, and advocacy on a range of policies and interventions, including regulatory policy, service and training expansion, professional scope of practice, and healthcare access and quality. PMAs in India derive and deploy their power in multiple ways, drawing on networks, professional authority, expertise and socioeconomic status to gain proximity to policy stakeholders or vocalize their concerns and thereby shape policy processes in their interests [22]. It is however important to recognize that PMAs are not homogenous; they represent diverse constituencies, and the roles and ideologies of organizations have evolved and varied over time. Despite their varied agendas, their role and influence among the medical community, health policy processes, and the wider health ecosystem carries high significance in India- as is the case globally [21].

In India, women have consistently endeavoured to establish and assert their collective voice within the medical field, specifically to address the healthcare needs of women. One of the earliest examples dates back to 1907, when the Association of Medical Women in India was founded with the objective of advocating for improved working conditions and fair compensation for women in medicine [23]. Subsequently, in 1928, the Indian Medical Association

(IMA) was formed as the first PMA representing physicians practicing modern medicine [24], with dedicated "wings" for women members. However, beyond this historical background, we have a very limited empirical understanding of the diversity in leadership within PMAs, including gender representation. Further, while limited data is available regarding the proportion of women in medical specializations, available information suggests low levels of representation [25, 26]. The current literature on demographics of PMA leadership is largely from high-income countries (HICs) and it suggests skewed gender representation in PMAs globally. For instance, globally in diabetes and endocrinology societies, 31.3% of board members were women [27], while female neurosurgeons held 29% of executive committee positions [28]. In Australian diabetes and endocrinology societies, women constituted 39% of council members [29], and only 10 of the 46 medical societies in Spain had a woman president between 2019–2021 [30]. The Canadian Medical Association has 20 men and six women in their board of directors [31], while out of 38 PMAs in the United States, 82.6% of the Presidents were men between 2008–2017 [32]. To the best of our knowledge, analyses of gender representation in professional medical associations in low- and- middle- income countries (LMICs) have not been published in the peer-reviewed literature. While published literature is unavailable, anecdotal evidence suggests that leadership positions in PMAs in many LMICs are dominated by men [33]. This lack of evidence leads to a gap in understanding the burden of the issue, further constraining the formulation of targeted strategies to rectify disparities. Bridging this gap is critical to advancing global gender equity, incorporating diverse perspectives into decision-making, and fortifying the effectiveness of healthcare systems.

In this study, we aim to analyse the gender composition of the current and past leadership of selected PMAs in India. We reviewed the gender composition of current and past leadership of PMAs ranging from the primary national medical association to associations representing formally recognized specialties. We conclude with policy and practice implications emerging from this research.

## Methods

In order to develop a list of PMAs for analysis, we identified general medical and physician-dominated public health associations, such as the IMA and the Indian Public Health Association (IPHA), and medical associations of all broad medical and surgical specialties as per the list of M.D. (Doctor of Medicine) and M.S. (Master of Surgery) post-graduation courses approved by the National Medical Commission (NMC) of India [34]. There were 29 broad medical and six surgical specializations in the NMC list, all of which, except Marine Medicine, were included. Marine Medicine has an association (Marine Medical Society of India), but we did not include it since it has very few members and is mostly involved in Navy warship training and has little engagement in public health or policy. Out of the others, 11 specialties had multiple professional associations, among which we selected the longest standing one for the analysis. Further, although Hospital Administration and Health Administration are separate MD specializations, they had a common PMA which was included in the analysis. In our final list, 33 broad medical specialty PMAs were included.

In addition to MD and MS post-graduation courses, there are super specialization courses for advanced specialised medical training in India. These associations are perceived to hold considerable power and influence in shaping health policy in the country. We also identified selected super-speciality associations that represent major medical and surgical fields. For this group, we selected PMAs of six medical specialities (Indian Academy of Neurology, Cardiological Society of India, Indian Society of Gastrology and Hepatology, Endocrine Society of India, Indian Society of Nephrology, and National Neonatalogy Forum) and five surgical super

specialities (Indian Association of Cardiovascular-Thoracic Surgeon, Indian Association of Gastrointestinal Endosurgeons, Urological Society of India, Indian Association of Surgical Oncology, and Neurological Society of India).

## Data extraction and analysis

Drawing upon methodologies used by Waseem et al and Sidhu et al. [27, 35], we used publicly available information from the official websites of these PMA as our primary source of data. We extracted the following data—year of inception, gender of the current president, gender of the current leadership committee members, and gender of all past presidents. The cut-off date for data extraction was 25th May 2023.

In this study, we adopted a binary construct of gender. To ascertain a person's gender, we used methodology that has been used in recent studies of gender composition analysis of health leadership [27, 35]. A sequential approach (i.e., moving to the next criteria if the previous one did not provide sufficient information) was adopted. First, it was checked if the PMA website included pictures accompanied by the name of the person and/or a gender-specific pronoun for members in the leadership committee. If it did not, the presence of gender signifiers such as Mrs, Ms, Miss, or Smt (Smt or Shreemati is commonly used as a title for a married woman in India) was noted. If neither of these were present, we used peer debriefing to ascertain gender- whether it was traditionally associated with women or men. In the case of a gender-neutral name, an internet search of the individual was also undertaken to determine if other sources of information were available regarding their gender. For the small number of people who had no internet presence, or their gender could still not be determined, the person was assumed to be a man, based on the rationale that women typically have gender signifiers in front of their names and that the field of medicine is overwhelmingly male.

The composition of the leadership committee across associations was not uniform, as some associations had zonal, state, or chapter-level leadership included in their central governing body, whilst others did not. We included data of the Central Committee as per the list of members mentioned in the official websites. For instance, the Association of Physicians of India includes leadership at the zonal level, ex-officio members, etc. in its Central Committee so all of them were included in our analysis. On the other hand, the Indian Association of Gastrointestinal Endosurgeons also has zonal-level leadership, however, they are not included in their Central Committee, and so were not included in our analysis.

The IMA is the largest PMA in India, with a membership of around 350,000 modern medicine doctors. To allow for decentralized decision-making, it has sub-chapters in all 28 states and four Union Territories of India. These institutions play a critical role in shaping medical practices and dictating healthcare policies and priorities within their respective regions. In addition to the above, we also analysed the leadership of the sub-chapters of IMA by extracting data on the gender of the presidents and secretaries. The complete dataset of extracted information has been provided as S1 Data.

This study did not require approval through an ethics committee as we have evaluated publicly available data. A reflexivity statement has been provided as S1 Text.

## Results

In total, 46 Indian PMAs were included, representing a mix of general, medical, and surgical specialities and super-specialities (Table 1). PMAs have a long history in India, with the IMA being the oldest (95 years) and the Indian Society of Sports and Exercise Medicine as the youngest (4 years). Out of 46 PMAs, 24 have been in existence since more than 50 years, while the average age of the included PMAs is 49 years.

**Table 1. Professional medical associations included in the study.**

| S. No. | Name of Professional Association | Practice speciality | Year of inception | Gender of current leadership |
|---|---|---|---|---|
| | **General** | | | |
| 1 | Indian Medical Association (IMA) | Modern medicine (general) | 1928 | M |
| 2 | Indian Public Health Association (IPHA) | Public Health | 1956 | M |
| | **MD Specializations** | | | |
| 1 | Indian Society of Anaesthesiologists (ISA) | Anesthesiology | 1947 | M |
| 2 | Indian Society of Aerospace Medicine (ISAM) | Aerospace Medicine | 1952 | F |
| 3 | Anatomical Society of India (ASI) | Anatomy | 1950 | F |
| 4 | Association of Clinical Biochemists of India | Biochemistry | 1975 | F |
| 5 | Indian Association of Preventive and Social Medicine (IAPSM) | Community Medicine | 1974 | M |
| 6 | Indian Association of Dermatologists, Venereologists and Leprologists (IADVL) | Dermatology, Venerology and Leprosy | 1973 | F |
| 7 | Academy of Family Physicians of India (AFPI) | Family Medicine | 2010 | M |
| 8 | Indian Academy of Forensic Medicine (IAFP) | Forensic Medicine | 1972 | M |
| 9 | Association of Physicians of India (API) | General Medicine | 1944 | M |
| 10 | Indian Academy of Geriatrics (IAG) | Geriatrics | 2002 | M |
| 11 | Academy of Hospital Administration (AHA) | Hospital & Health Administration | 1987 | M |
| 12 | Indian Society of Transfusion Medicine (ISTM) | Immuno-Hematology and Blood Transfusion | 2011 | M |
| 13 | Indian Association of Medical Microbiology (IAMM) | Microbiology | 1976 | F |
| 14 | Society of Nuclear Medicine in India (SNMI) | Nuclear Medicine | 1967 | M |
| 15 | Indian Association of Pathologists and Microbiologists (IAPM) | Pathology | 1949 | F |
| 16 | Indian Association of Pediatrics (IAP) | Pediatrics | 1963 | M |
| 17 | Indian Pharmacological Society (IPS) | Pharmacology | 1966 | M |
| 18 | Indian Association of Physical Medicine and Rehabilitation (IAPMR) | Physical Medicine Rehabilitation | 1972 | M |
| 19 | Association of Physiologists of India (ASSOPI) | Physiology | N/A | F |
| 20 | Indian Psychiatric Society (IPS) | Psychiatry | 1947 | M |
| 21 | Indian Radiology and Imaging Association (IRIA) | Radio-diagnosis | 1931 | M |
| 22 | Association of Radiation Oncologists of India (AROI) | Radiation Oncology | 1992 | M |
| 23 | Indian Society of Sports and Exercise Medicine (ISSEM) | Sports Medicine | 2019 | M |
| 24 | Society of Tropical Medicine & Infectious Diseases in India (STMID) | Tropical Medicine | 2009 | M |
| 25 | Indian Chest Society (ICS) | Respiratory Medicine | 1980 | M |
| 26 | Society for Emergency Medicine India (SEMI) | Emergency Medicine | 1999 | M |
| 27 | Indian Association of Palliative Care (IAPC) | Palliative Medicine | 1984 | F |
| | **MS Specializations** | | | |
| 1 | Association of Otolaryngologists of India (AOI) | Otorhinolaryngology | 1947 | F |
| 2 | Association of Surgeons in India (ASI) | General Surgery | 1983 | M |
| 3 | All India Ophthalmological Society (AIOS) | Ophthalmology | 1930 | M |
| 4 | Indian Orthopaedic Society (IOA) | Orthopedics | 1955 | M |
| 5 | Federation of Obstetric and Gynaecological Societies of India (FOGSI) | Obstetrics & Gynaecology | 1950 | M |
| 6 | Indian Society For Trauma And Acute Care (ISTAC) | Traumatology and Surgery | N/A | M |
| | **Select super specialities—Medical** | | | |
| 1 | Indian Academy of Neurology (IAN) | Neurology | 1992 | M |
| 2 | Cardiological Society of India (CSI) | Cardiology | 1946 | M |
| 3 | Indian Society of Gastrology and Hepatology (ISGH) | Gastroenterology | N/A | M |
| 4 | Endocrine Society of India (ESI) | Endocrinology | 1971 | M |

*(Continued)*

**Table 1.** (Continued)

| S. No. | Name of Professional Association | Practice speciality | Year of inception | Gender of current leadership |
|---|---|---|---|---|
| 5 | Indian Society of Nephrology (ISN) | Nephrology | 1970 | M |
| 6 | National Neonatalogy Forum (NNF) | Neonatology | 1980 | M |
| | **Select super specialities—Surgical** | | | |
| 1 | Indian Association of Cardiovascular-Thoracic Surgeon (IACTS) | Cardiothoracic Surgery | 1990 | M |
| 2 | Indian Association of Gastrointestinal Endosurgeons (IAGES) | GI surgery | 1993 | M |
| 3 | Urological Society of India (USI) | Urology | 1961 | M |
| 4 | Indian Association of Surgical Oncology (IASO) | Oco-surgery | 1977 | M |
| 5 | Neurological Society of India (NSI) | All neuro Specialities | 1951 | M |

## Current leadership

Of the 46 PMAs, nine (19·5%) were led by a woman president. These include the Indian Society of Aerospace Medicine, Anatomical Society of India, Association of Clinical Biochemists of India, Indian Association of Dermatologists, Venereologists, and Leprologists, Indian Association of Medical Microbiology, Indian Association of Pathologists and Microbiologists, Indian Association of Palliative Care, Association of Physiologists of India, and Association of Otolaryngologists of India. None of the PMAs representing the selected super specialities (surgical or medical) were led by a woman.

## Current leadership committee

In total, there were 659 members in the current leadership committees of the 46 PMAs, out of which 106 (16%) were women. Nine PMAs have no women in their leadership committees-five of these being super speciality associations. Only six PMAs have a 50% or higher representation of women in their current central leadership committee; these are Indian Association of Palliative Care (100%), Federation of Obstetric and Gynaecological Societies of India (61·5%), Indian Association of Pathologists and Microbiologists (57%), Indian Association of Dermatologists, Venereologists and Leprologists (55·5%), Indian Association of Medical Microbiology (50%), and Academy of Family Physicians of India (50%) (Fig 1).

## Past leadership

Of the 46 PMAs, historical information on leadership was publicly available for 31 associations. These 31 organizations have been led by a total of 1388 presidents till date, out of which only 78 (5·6%) were women. Individually, all 31 associations had equal to or less than 20% women presidents, and 24 have had equal to or less than 10% women presidents in their history. Five associations- the Indian Society of Aerospace Medicine, Academy of Hospital Administration, Indian Orthopaedic Society, Indian Association of Gastrointestinal Endosurgeons, and the Urological Society of India- have had no women presidents since their inception, in their average history of 53 years.

Eight PMAs have had more than 10% of women presidents in past, these are- Indian Association of Medical Microbiology (20%), Society for Emergency Medicine India (18.1%), National Neonatalogy Forum (16·6%), Indian Association of Preventive and Social Medicine (16.6%), Federation of Obstetric and Gynaecological Societies of India (15%), Association of Clinical Biochemists of India (12·7%), Indian Academy of Geriatrics (12·5%), and Indian Society of Anaesthesiologists (10%) (Fig 2).

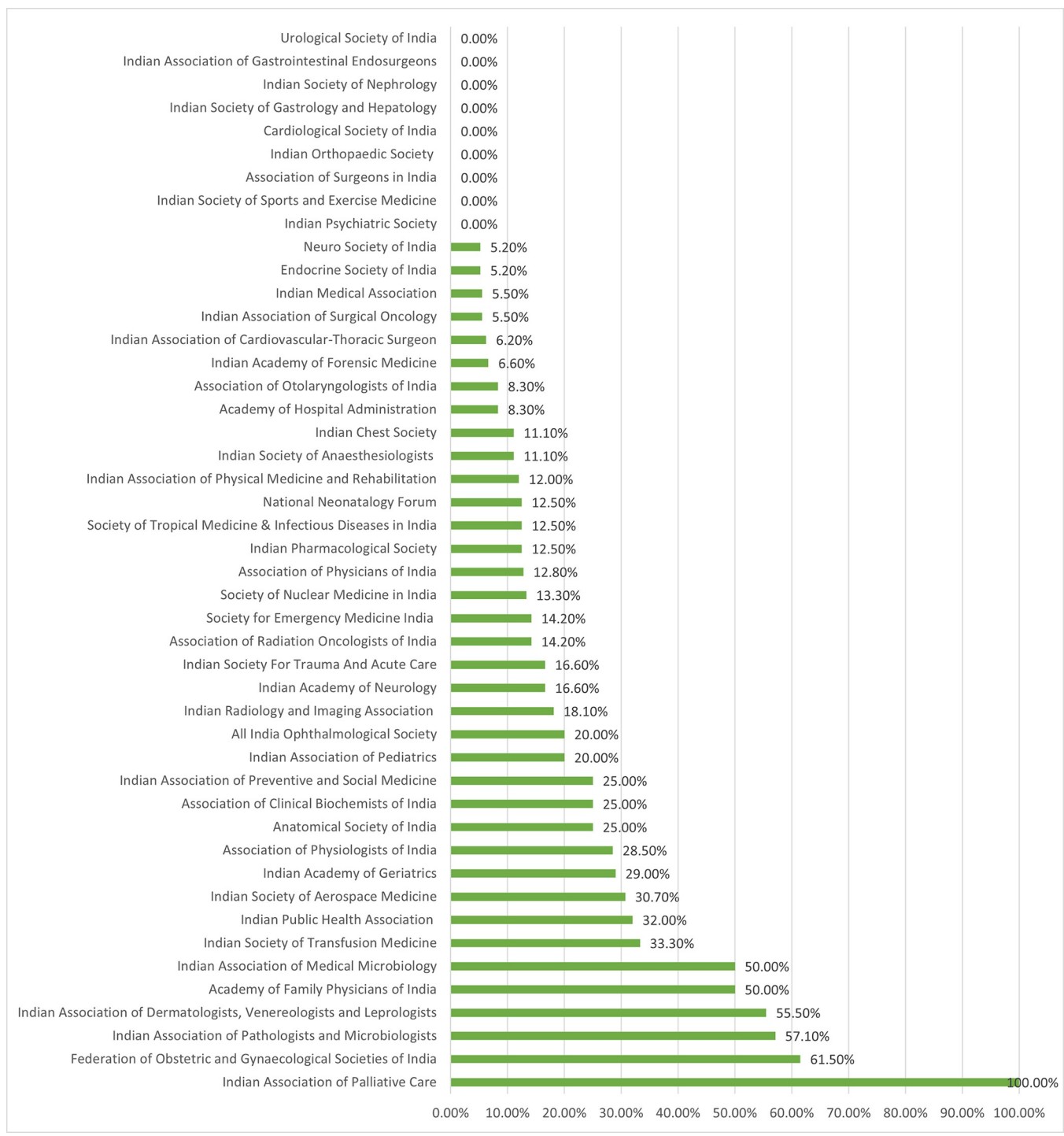

**Fig 1. Proportion of women in current central committees of professional medical associations.**

## The leadership of the Indian Medical Association (IMA)

Established in 1928, prior to the country's independence, the IMA boasts a long-standing history, with the largest voluntary organization of modern medicine doctors in India. An election process decides the association's leadership for one or two years [24]. However, the

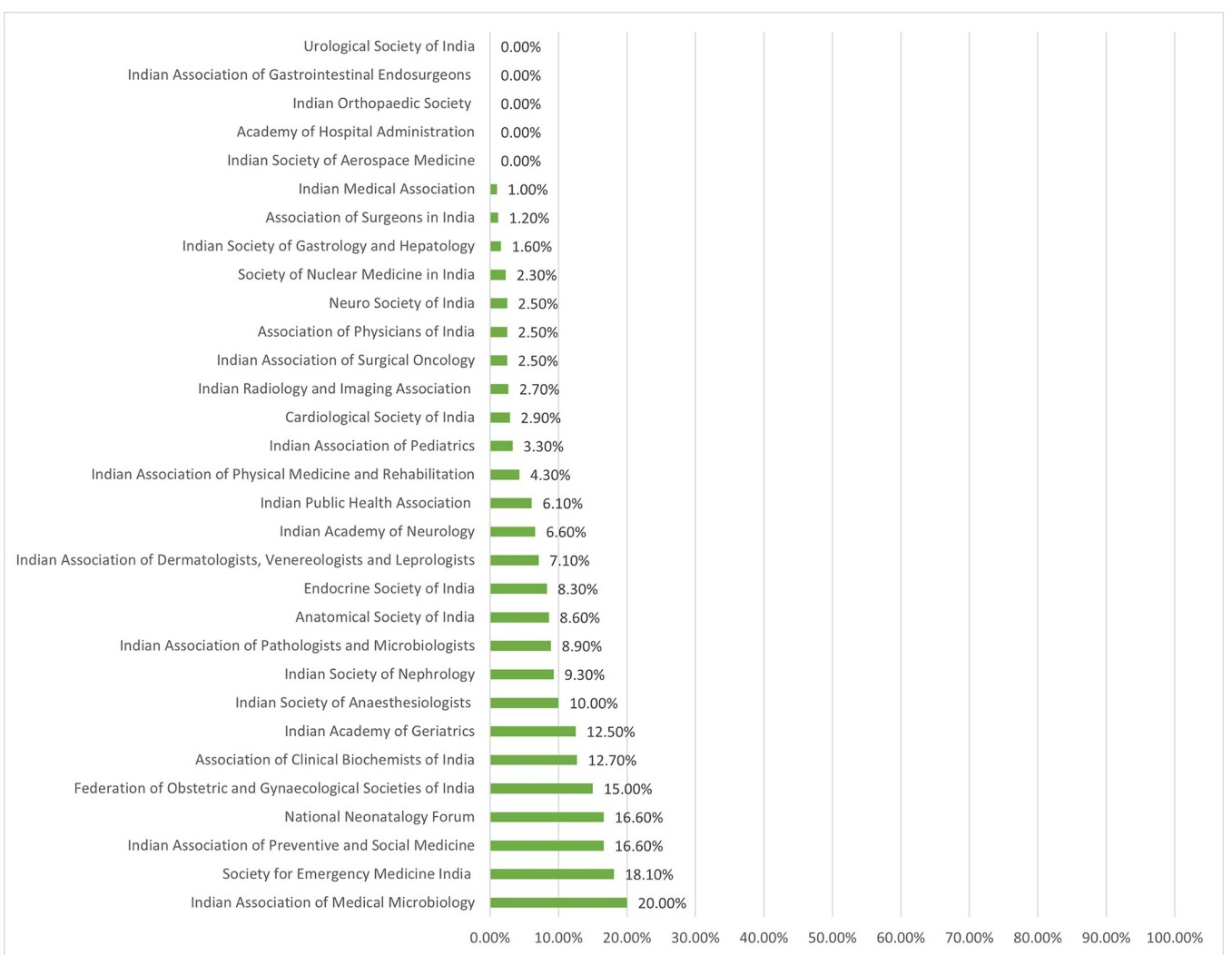

**Fig 2. Proportion of past women presidents of professional medical associations.**

representation of women in leadership roles within the IMA remains negligible. The association is currently led by a man and the current governing body consists of 18 members, of which only one is a woman, accounting for 5·5% representation. Further, out of the 92 individuals who have served as presidents since its inception, only one was a woman.

In addition, of the 64 individuals that are currently serving as presidents and secretaries of the 32 state sub-chapters of IMA, only three are women. The state chapter in Haryana has a woman in both president and secretary positions, while the state of Goa has a woman as secretary. All the remaining 61 positions are occupied by men. Moreover, as per the IMA website, they have a Women Doctor's Wing, which exists nationally and at the state chapter level to 'look after the interests of women doctors,' but the details are not publicly available, and their role in organizational governance is not clear.

## Discussion

This study examined the representation of women in leadership of PMAs in India. The findings reveal minimal representation of women, both currently and historically. To the best of

our knowledge, this is the first attempt to decipher the gender composition of PMAs in India and LMICs. The relevance of these findings extends beyond the Indian and LMIC context, as evidence from HICs also reveal a comparable pattern of women facing challenges in attaining leadership positions in PMAs [27–29, 35]. These results are consistent with global studies indicating the 'leaky pipeline'- that although there are more women entering the medical field than ever before, very few reach leadership [1, 8]. This underrepresentation is linked to prevalent gender stereotypes and discrimination and unconscious favouring of men in decision-making and leadership roles. Further, social and cultural expectations related to parental responsibilities and caregiving such as the pressure to prioritize family responsibilities over career advancement, limited support for work-life balance, and societal norms that undervalue women's professional contributions, significantly contribute to the barriers women face in advancing their careers and pursuing leadership opportunities [1, 2]. PMAs also operate within the inherently technocratic and gendered structure of the medical field, a system primarily designed by and for traditional gender roles associated with men [36]. Considering the predominant role of PMAs in shaping medical training and health policies in India and beyond, skewed gender representation is a major concern. Moreover, India ranks 122 out of 170 countries in the Gender Inequality Index [37], indicating severe gaps in equity in the country overall, a fact that further grounds and contextualizes the lack of gender representation in Indian PMAs reported in this study.

Considering the predominant role of PMAs in shaping medical training, practice, and their influence on health policies in India and beyond, the gross underrepresentation of women is a major concern. The IMA is the oldest, largest, and most prominent PMA in India, with historical contributions in influencing health policies in the country. The systemic exclusion of women from such associations restricts their voice and their unique perspectives and experiences from the decision-making processes. This lack of diverse representation can lead to an unbalanced agenda that may not adequately address the health and care needs of women or sufficiently take into account gender-specific issues when advocating for or collaborating on particular health policy and programs. Moreover, despite the process of selection of leadership committees through an election by the members themselves, gender representation continues to remain dismal. This phenomenon may be attributed to unconscious gender bias [38], indicating the need for further investigation of the barriers preventing women from ascending to top leadership positions in PMAs.

Our results show associations related to women's and children's health, such as obstetrics and gynaecology, paediatrics, and neonatology, also have unequal representation. For instance, the National Neonatology Forum has only one woman in its leadership committee and in FOGSI's 73 years of history, only 15% of past presidents were women, highlighting the issue of men occupying leadership positions in such fields. This is not to suggest that women and children's health should exclusively be led by women; rather, we argue for better gender representation to align leadership with the organization's membership and to foster more inclusive and equity-driven decision-making, leveraging the contributions of women leaders in these fields. Current trends in gynaecology suggest that the majority of practicing physicians are women [39], yet the continued dominance of men in leadership positions indicates the presence of a glass ceiling for women. This dominance by men in women's health has implications for the prioritization and focus of medical practice and policy. As noted by Morgan et al., gendered power relations affect "the health workforce (whether informal care provided at home is recognised and supported; whether recruitment, retention, promotion and harassment policies take gender bias into consideration), health financing (the extent of financial protection availability to different groups, out-of-pocket expenditures) and governance (the systems of daily management, leadership, accountability and the extent to which policies incorporate gender

considerations)" [40]. Evidence also highlights that the overwhelming presence of men in the field of women's health contributes to an excessive emphasis on women's reproductive roles rather than addressing their holistic health needs [41]. Moreover, an emerging body of research highlights violence and discrimination experienced by women physicians in India, an issue that greater representation of women in leadership positions can contribute to addressing [42].

Our study underscores the urgent need for concerted efforts to address gender inequities in leadership of PMAs in India. It is evident that having women-only chapters within associations is insufficient in addressing this issue effectively since despite their presence in the IMA, only 5.5% of its leadership board is women. This signifies the need to go beyond mere inclusion and implement a gender-transformative leadership approach by including affirmative action policies and initiatives that promote equal opportunities for women and other genders in the medical profession. Strategies such as setting targets and quotas, mentorship programs, and leadership training can foster sensitivity and inclusion, leading to diverse representation in leadership positions. It is also imperative to create a supportive work environment and culture by including childcare facilities, flexible work timings, promoting female role models, strong guidelines against sexual harassment, safe transportation facilities, and transparent leader selection and promotion [36]. In addition, it is important to minimize the gender pay gap and provide fair and equal compensation to all employees for the same levels of work [4]. Lastly, fostering awareness and advocacy for gender equality within the medical community can help support the necessary cultural and systemic changes required for achieving equitable representation in leadership roles in the health workforce and beyond [2, 3, 36]. Having such measures in place will contribute significantly towards accomplishing the Sustainable Development Goals, particularly Goal 3 of Good Health and Wellbeing and Goal 5 of Gender Equality. We recommend that further studies explore barriers and facilitators to women's leadership in PMAs in India, particularly through qualitative measures.

Further, it is important to note that advocating for women's representation in leadership is just the tip of the iceberg and should not become a checkbox activity. PMAs must have an intersectional lens to diversity and inclusion, with respect to gender, socio-economic status, inclusion of marginalized and minority groups, geographic diversity and more. Wider societal changes are also needed to promote equitable policies, recognizing that achieving more equitable leadership is just one piece of the broader puzzle. These changes should encompass shifts in cultural norms, institutional practices, and policy frameworks that collectively foster a more inclusive and equitable environment for all genders. This holistic approach is needed to dismantle the existing barriers and ensure lasting gender parity across the medical profession and within leadership positions.

## Limitations

Our analysis presents a novel and unique contribution by shedding light on the significant gender disparities within Indian PMAs. However, it is important to acknowledge and discuss some limitations associated with our study. All the data collected for analysis were sourced from publicly accessible websites, and in some associations, the year for their governing bodies was not specified. Thus, we have relied on the assumption that the information provided was correct and updated. In addition, while we included PMAs from all broad specializations, we only selected certain super-specialty associations based on their engagement in policy and practice. Therefore, though our findings can be generalized at the local and national level to the broad specialty categories, but not to super-specialties.

It is also possible that the method employed to determine gender may have resulted in mis-attributions since gender identification based solely on indices such as pronouns and images can be subject to inaccuracies. Further, our analysis could not adopt an intersectional lens due to the gender data gap, necessitating us to 'ascertain' gender based on publicly available signifiers. Where gender signifiers were not present, we assumed the person to be a man; however, this was only 3% of our total sample. We have also adopted a binary construct of gender due to the gap in available data, which may fail to encompass the intricacies and diversity of other gender identities. In an effort to mitigate these limitations, we made efforts to minimize biases by adopting a sequential multi-step methodology that has been used in similar recent studies [27, 35]. By following established practices, we aimed to enhance the reliability and consistency of gender identification in our analysis. Nonetheless, it is crucial to recognize that despite our efforts, some degree of misclassification and unintentional exclusion of gender-diverse and non-binary individuals may have occurred.

## Conclusion

The representation of women in the leadership positions of professional medical associations in India highlights significant gender disparities. Given that diverse leadership brings unique perspectives and enhances decision-making, PMAs can lead by example by implementing affirmative measures to empower women and ensure equal representation in leadership positions. This would help harness the full potential of women and contribute to the development of a more inclusive and gender-equitable healthcare system and policies.

## Supporting information

**S1 Data. Dataset.**
(XLSX)

**S1 Text. Reflexivity statement.**
(DOCX)

## Acknowledgments

The authors would like to thank and acknowledge the contribution of Dr Kaveri Mayra for comments on an earlier draft of the manuscript.

## Author Contributions

**Conceptualization:** Veena Sriram, Vikash R. Keshri.

**Data curation:** Pratishtha Singh.

**Formal analysis:** Pratishtha Singh, Vikash R. Keshri.

**Investigation:** Veena Sriram, Sonali Vaid, Vikash R. Keshri.

**Methodology:** Pratishtha Singh.

**Project administration:** Pratishtha Singh.

**Supervision:** Veena Sriram, Vikash R. Keshri.

**Writing – original draft:** Pratishtha Singh.

**Writing – review & editing:** Veena Sriram, Sonali Vaid, Sharmishtha Nanda, Vikash R. Keshri.

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
