## [Decision Letter · Decision Letter 0]

13 May 2024

PGPH-D-23-02642

Examining representation of women in leadership of professional medical associations in India

Dear Dr.

Thank you for submitting your manuscript to PLOS Global Public Health. After careful consideration, we feel that it has merit but does not fully meet PLOS Global Public Health’s publication criteria as it currently stands. Therefore, we invite you to submit a revised version of the manuscript that addresses the points raised during the review process.

We look forward to receiving your revised manuscript.

Kind regards,

Nazik Hammad, MD, FACP, FRCPC

Academic Editor

Journal Requirements:

Additional Editor Comments (if provided):

Reviewers' comments:

Reviewer's Responses to Questions

**Comments to the Author**

1. Does this manuscript meet PLOS Global Public Health’s publication criteria? Is the manuscript technically sound, and do the data support the conclusions? The manuscript must describe methodologically and ethically rigorous research with conclusions that are appropriately drawn based on the data presented.

Reviewer #1: Yes

Reviewer #2: Yes

Reviewer #3: Yes

Reviewer #4: Yes

2. Has the statistical analysis been performed appropriately and rigorously?

Reviewer #1: Yes

Reviewer #2: Yes

Reviewer #3: Yes

Reviewer #4: N/A

3. Have the authors made all data underlying the findings in their manuscript fully available (please refer to the Data Availability Statement at the start of the manuscript PDF file)?

Reviewer #1: Yes

Reviewer #2: Yes

Reviewer #3: Yes

Reviewer #4: Yes

4. Is the manuscript presented in an intelligible fashion and written in standard English?

Reviewer #1: Yes

Reviewer #2: Yes

Reviewer #3: Yes

Reviewer #4: Yes

5. Review Comments to the Author

Reviewer #1: This is an important contribution to human resources in health. It would be useful if the authors indicate the proportion of women in the major specialisations. This would help to support their argument that women are under represented in certain specialisations and over represented in others. How does this get reflected in their presence in professional organisations?

Reviewer #2: Strengths

1. strong arguments

2. well written

weakness

1. long introduction which can be shorten

2, lots of repetitive arguments and statements

3. will recommend a graph indicating the changes over the decades

4. Discussion should include cultural and social roles of women what could be contributing to a leaky pipeline

5. any comparison with any other Asean country data, allied health organizations including nursing and radiographers?

Reviewer #3: A very well written manuscript presenting important data with an excellent discussion in context.

Limitations are discussed however suggest to expand discussion of the sampling method of the societies reviewed - how did the authors ensure that the societies in the sample are representative and how confident are they that the results can be generalized to all local/national societies

Reviewer #4: I applaud the authors on this insightful manuscript with addresses gender disparities in leadership from a LMIC perspective.

Suggest the authors to elaborate on the 32 sub chapters of the Indian Medical Association to provide more context to readers who are familiar with the organizational structures of Indian PMAs.

May the authors justify in the methodology why an assumption was made regarding gender for this specific study. ( When gender is not identified, assumption was male by default.

Line 169- Is this meant to be 350 000 or 3 500 000

Line 179- There is reference to lifespan of associations - did some dissolve? If not, suggestion is to rephrase to make it clearer.

The authors provided results of current and past leadership which is a very important aspect of the study. Considering the oldest association is 95years, is it possible to review the trend for each decade regarding the number of women in leadership. It will be enlightening to note if overall there is an improvement or rather stagnation to guide interventions that assist in promoting women participation in leadership roles overall.

Do we also have the number of women who are members in some of the associations referenced specifically from India. They could use data for IMA only for example, or obstetric and gynecology association to obtain the proportion of women who are members.

Overall, an important and crucial topic that overall can improve health outcomes in this population .

6. PLOS authors have the option to publish the peer review history of their article (what does this mean?). If published, this will include your full peer review and any attached files.

**Do you want your identity to be public for this peer review?** For information about this choice, including consent withdrawal, please see our Privacy Policy.

Reviewer #1: No

Reviewer #2: **Yes: **verna vanderpuye

Reviewer #3: No

Reviewer #4: No

---

## [Decision Letter · Decision Letter 1]

19 Jul 2024

Examining representation of women in leadership of professional medical associations in India

PGPH-D-23-02642R1

Dear Dr. Pratishtha Singh,

We are pleased to inform you that your manuscript 'Examining representation of women in leadership of professional medical associations in India' has been provisionally accepted for publication in PLOS Global Public Health.

Best regards,

Nazik Hammad, MD, FACP, FRCPC

Academic Editor

Reviewer Comments (if any, and for reference):

Reviewer's Responses to Questions

**Comments to the Author**

1. If the authors have adequately addressed your comments raised in a previous round of review and you feel that this manuscript is now acceptable for publication, you may indicate that here to bypass the “Comments to the Author” section, enter your conflict of interest statement in the “Confidential to Editor” section, and submit your "Accept" recommendation.

Reviewer #3: All comments have been addressed

Reviewer #4: (No Response)

2. Does this manuscript meet PLOS Global Public Health’s publication criteria? Is the manuscript technically sound, and do the data support the conclusions? The manuscript must describe methodologically and ethically rigorous research with conclusions that are appropriately drawn based on the data presented.

Reviewer #3: Yes

Reviewer #4: Yes

3. Has the statistical analysis been performed appropriately and rigorously?

Reviewer #3: N/A

Reviewer #4: N/A

4. Have the authors made all data underlying the findings in their manuscript fully available (please refer to the Data Availability Statement at the start of the manuscript PDF file)?

Reviewer #3: Yes

Reviewer #4: Yes

5. Is the manuscript presented in an intelligible fashion and written in standard English?

Reviewer #3: Yes

Reviewer #4: Yes

6. Review Comments to the Author

Reviewer #3: All comments have been addressed

Reviewer #4: The introduction can be further summarized . Suggest retaining first paragraph as is. Paragraph 2 , authors may consider editing and in cooperating the following lines 50;51;53-55;68-70;73-81;87-89. Third and last paragraph can be lines 120-123.

The edited out content is important information that may be used to support results in the discussion section.

Otherwise, well written and an important study .

7. PLOS authors have the option to publish the peer review history of their article (what does this mean?). If published, this will include your full peer review and any attached files.

**Do you want your identity to be public for this peer review?** For information about this choice, including consent withdrawal, please see our Privacy Policy.
